# Fecal Luminal Factors from Patients with Gastrointestinal Diseases Alter Gene Expression Profiles in Caco-2 Cells and Colonoids

**DOI:** 10.3390/ijms232415505

**Published:** 2022-12-07

**Authors:** Luiza Moraes Holst, Cristina Iribarren, Maria Sapnara, Otto Savolainen, Hans Törnblom, Yvonne Wettergren, Hans Strid, Magnus Simrén, Maria K. Magnusson, Lena Öhman

**Affiliations:** 1Department of Microbiology and Immunology, Institute of Biomedicine, Sahlgrenska Academy, University of Gothenburg, 405 30 Gothenburg, Sweden; 2Chalmers Mass Spectrometry Infrastructure, Department of Biology and Biological Engineering, Chalmers University of Technology, 412 96 Gothenburg, Sweden; 3Institute of Public Health and Clinical Nutrition, University of Eastern Finland, 70210 Kuopio, Finland; 4Department of Molecular and Clinical Medicine, Institute of Medicine, Sahlgrenska Academy, University of Gothenburg, 405 30 Gothenburg, Sweden; 5Department of Surgery, Institute of Clinical Sciences, Sahlgrenska University Hospital, University of Gothenburg, 405 30 Gothenburg, Sweden; 6Department of Internal Medicine, Södra Älvsborgs Hospital, 501 82 Borås, Sweden

**Keywords:** host–microbial crosstalk, organoids, Caco-2, colon cancer, ulcerative colitis, irritable bowel syndrome, fecal metabolites, gut barrier

## Abstract

Previous in vitro studies have shown that the intestinal luminal content, including metabolites, possibly regulates epithelial layer responses to harmful stimuli and promotes disease. Therefore, we aimed to test the hypothesis that fecal supernatants from patients with colon cancer (CC), ulcerative colitis (UC) and irritable bowel syndrome (IBS) contain distinct metabolite profiles and establish their effects on Caco-2 cells and human-derived colon organoids (colonoids). The metabolite profiles of fecal supernatants were analyzed by liquid chromatography–mass spectrometry and distinguished patients with CC (*n* = 6), UC (*n* = 6), IBS (*n* = 6) and healthy subjects (*n* = 6). Caco-2 monolayers and human apical-out colonoids underwent stimulation with fecal supernatants from different patient groups and healthy subjects. Their addition did not impair monolayer integrity, as measured by transepithelial electrical resistance; however, fecal supernatants from different patient groups and healthy subjects altered the gene expression of Caco-2 monolayers, as well as colonoid cultures. In conclusion, the stimulation of Caco-2 cells and colonoids with fecal supernatants derived from CC, UC and IBS patients altered gene expression profiles, potentially reflecting the luminal microenvironment of the fecal sample donor. This experimental approach allows for investigating the crosstalk at the gut barrier and the effects of the gut microenvironment in the pathogenesis of intestinal diseases.

## 1. Introduction

The intestinal epithelium consists of a dynamic single layer of intestinal epithelial cells (IEC) that mediate the interaction between the host and the local intestinal microenvironment. Aside from nutrient absorption, the epithelium is an integral part of the innate immune system, providing a physical barrier to noxious substances and pathogens, while actively resisting microbial invasion by secreting mucus, antimicrobial peptides and hormones [1]. The intestine, and more specifically the epithelial barrier, is central for the pathogenesis of several diseases, such as colon cancer (CC), ulcerative colitis (UC) and irritable bowel syndrome (IBS) [2,3,4]. Evidence suggests that the impairment of the epithelial barrier integrity, called “leaky-gut”, might allow translocation of the luminal microbiota and its metabolites, triggering mucosal immune responses and inflammation [5].

Whether imbalance of the intestinal microbiota composition is the initial trigger or the result of inflammation is unclear. Indeed, epithelial barrier defects, modified expression of antimicrobial peptides, imbalanced gut microbiota and altered fecal metabolite profiles have all been reported in patients with CC, UC and IBS, indicating a dysfunctional intestinal barrier for disease development [6,7]. Furthermore, chronic intestinal inflammation and epithelial damage are known to affect microbial composition and are linked to intestinal carcinogenesis [8,9]. However, the mechanisms behind IEC dysfunction, crosstalk with the gut microbiome and the communication with the immune system are yet to be fully understood.

Regarding the gut barrier as an interface exposed to microbiota and the luminal content, the functional and structural characteristics of the epithelium have been widely studied with the help of immortalized cell lines, such as the colon carcinoma-derived Caco-2 cells. In vitro models that investigate the effects of stimulating cell line cultures with specific microbial metabolites, food nutrients or cytokines are the most common. Studies that use Caco-2 monolayer systems show that microbial metabolites exert both inflammatory [10] and protective effects [11], suggesting that the microbiota can modulate the epithelial response to harmful stimuli. However, Caco-2 cells can only differentiate into enterocyte-like cells that represent some of the specialized IEC types. Thus, although extensively used for research, they constitute an over-simplified model to study gut barrier interactions. Therefore, human-derived organoids have emerged as an attractive in vitro system to study the intestinal microenvironment. Intestinal organoids are generated from tissue-resident stem cells collected from biopsies and differentiate into the various specialized epithelial cells during specific conditions, forming a more complete version of the intestinal border than the Caco-2 cells [12,13,14,15,16]. The use of colon organoids derived from CC and UC patients for studying disease mechanisms and developing individualized therapy strategies has evolved during recent years [17,18,19,20]. However, studies that investigate the effects of the microbiota on human-derived intestinal organoids are limited [21,22,23], and it needs to be determined if external stimuli, such as components and metabolites found in the lumen, are able to induce disease specific phenotypes in organoids derived from healthy human biopsies.

In this study, we aimed to test the hypothesis that fecal supernatants from patients with CC, UC and IBS contain distinct metabolite profiles. Furthermore, we aimed to establish the effects of patient-derived fecal supernatants, potentially reflecting the intestinal microenvironment of the donor, on traditional Caco-2 cell monolayers and human-derived colon organoids (colonoids).

## 2. Results

### 2.1. Study Subjects

In total, six patients with CC, six UC patients with active disease, six IBS patients and six healthy subjects provided fecal samples for the preparation of the fecal supernatants used in this study (for patient demographics, see Appendix A). Patients with CC were older than the healthy subjects (*p* < 0.01) and IBS patients (*p* < 0.05), but no other age differences were observed. Sex distribution was equal for all groups. All CC patients had a primary tumor located in the right colon, where two subjects were diagnosed with stage I, one subject with stage II, two subjects with stage III and one subject with stage IV colon cancer. For UC patients, the total Mayo score for disease activity had a median value of 9 [6,7,8,9,10,11], and four patients had an endoscopic score of 2 and two patients had a score of 3. Lastly, among the IBS patients, four patients had predominant diarrhea and two patients had mixed bowel habits with a median IBS-SSS score of 278 (200–377).

### 2.2. Metabolite Profiles from Fecal Supernatants Distinguish between Patient Groups and Healthy Subjects

To evaluate if the metabolite composition of the fecal supernatants was unique to each study group, an untargeted LC–MS analysis was performed, resulting in 9699 spectral features. A PCA that included all features comparing healthy subjects, patients with CC, UC and IBS revealed four separate clusters (Figure 1) with minimal overlap. Pairwise OPLS-DA analyses between each of the patient groups with the healthy subjects revealed that 413 spectral features contributed to the separation between CC and healthy subjects, 294 spectral features contributed to the separation between UC and healthy subjects, and lastly, 392 spectral features contributed to the separation between IBS and healthy subjects (Appendix A).

### 2.3. Fecal Supernatants from Patient Groups and Healthy Subjects Alter Gene Expression Profiles of Caco-2 Monolayer Cultures

To investigate the effects of luminal metabolites on epithelial cells, fecal supernatants were added to the cell culture medium in the apical chamber of transwell Caco-2 monolayers. A fecal supernatant dilution of 1:100 was used based on a titration assay where the lowest dose that induced a beneficial effect on Caco-2 cell metabolism was chosen (Appendix A). As controls of the transwell model, LPS was added to the apical chamber, TNF was added to the basolateral chamber or the cells were left untreated. TEER measurement was performed immediately prior to stimulation and after 48h of culture, a decrease in TEER for cells treated with fecal supernatants from CC patients was observed, while no other differences could be detected (Figure 2A). After 48 h of culture, cells were collected for gene expression analysis.

PCA based on gene expression data showed that cells cultured with fecal supernatants from healthy subjects clustered separately from untreated cells and from cells cultured in the presence of LPS or TNF (Figure 2B). A second PCA that included only fecal supernatant-stimulated Caco-2 cells indicated differences when stimulating cell cultures with fecal supernatants from healthy subjects and patients with CC, UC and IBS, but with overlap between the study groups (Figure 2C).

To define the most important genes that drive the differentiation between the groups, pairwise OPLS-DA analyses were performed to compare Caco-2 cells cultured in the presence of fecal supernatants from each patient group with healthy subjects. Analysis of the gene expression data in OPLS-DAs revealed *LYZ, IL1B, CLDN2, CCL20, TLR7, STAT3, BCL2*, and *CD1D* to be among the most important genes for differentiation between CC vs. healthy subjects (Figure 2D); *BIRC3, CLDN2* and *CD1D* for UC vs. healthy subjects (Figure 2E); and finally, *CLDN2, IL10RA1, DSC2, BIRC3, CX3CL1,* and *CD1D* for IBS vs. healthy subjects (Figure 2F).

Next, specific patterns induced by fecal supernatants from CC, UC and IBS patients on Caco-2 cell monolayers were assessed by comparing gene expression data in a PCA biplot (Figure 2G). Expression of pro inflammatory genes, including *CXCL1, CXCL2, CCL20* and *IL1B,* was increased when cell cultures were stimulated with fecal supernatants from UC and CC patients, when compared to fecal supernatants from IBS patients. In contrast, the epithelial barrier integrity gene, *OCLN*, and antimicrobial response gene, *TLR2*, showed higher expression in cell cultures stimulated with fecal supernatants from IBS patients, in relation to the other patient groups. Furthermore, expression of the other genes involved in antimicrobial response, including *JUN, IRF7* and *RELA,* was increased in cell cultures when stimulated with fecal supernatants from UC and IBS patients, in comparison with fecal supernatants from CC patients.

Last, we compared the expression between all study groups for the selected genes identified in Figure 2D–F. Fecal supernatants from CC patients induced higher gene expression of *IL1B* (Figure 3A), whereas fecal supernatants from patients with UC induced higher expression of *BIRC3* (Figure 3B)*,* when compared to healthy subjects. Fecal supernatants from patients with IBS induced higher expression of the barrier integrity-related genes *CLDN2* (Figure 3C) and *DSC2* (Figure 3D) than fecal supernatants from healthy subjects. In summary, addition of fecal supernatants to Caco-2 monolayers induced distinct gene expression patterns, with some patient group-specific differences.

### 2.4. Fecal Supernatants from Patient Groups and Healthy Subjects Alter Gene Expression Profiles of Human Colonoid Cultures

Next, we investigated the effects of fecal supernatants on apical-out colonoids (Appendix A). The supernatants were added into the culture media that contained suspended colonoids and LPS was included as a control, in addition to untreated wells. After 48 h of culture, cells were collected for gene expression analysis.

First, a PCA based on gene expression revealed a distinct profile for the colonoids cultured in the presence of fecal supernatants from healthy subjects when compared to the colonoids cultured with LPS or left untreated (Figure 4A). When comparing the gene expression of colonoids cultured in the presence of fecal supernatants from the various study groups, the PCA revealed overlapping clusters of cultures stimulated with supernatants from CC, UC and healthy subjects, while the colonoids stimulated with fecal supernatants from IBS patients tended to cluster separate from the other groups (Figure 4B).

To define the most important variables that differentiate between colonoids stimulated with fecal supernatants from healthy subjects and the different patient groups, pairwise analyses were performed based on gene expression. OPLS-DA analyses identified *IL12A, TNF,* and *TRAF6 to be* among the most important genes for discrimination between colonoids cultured in the presence of fecal supernatants from CC patients vs. healthy subjects (Figure 4C); *IL12A, TLR5, IRF7, TNF, IL10RA1, PTGS2, CXCL2* and *CCL2* for UC patients vs. healthy subjects (Figure 4D); and *IL12A, IL10RA1, IFNA1,* and *NOTCH1* for IBS patients vs. healthy subjects (Figure 4E).

A PCA biplot that compared gene expression colonoids stimulated with fecal supernatants from CC, UC and IBS patients revealed that the colonoids assumed distinct expression patterns (Figure 4F). Colonoids stimulated with fecal supernatants from CC and UC patients assumed a mixed gene expression pattern, with higher levels of *RELA, TP53, TRAF6, TNFSF13, DSC2* and *CASP8*. A second small cluster of colonoids stimulated with fecal supernatants from CC and UC showed higher expression of *IL8, ANXA1, NOS2, IL23A, BIRC3* and *ALOX12*. Furthermore, fecal supernatants from IBS patients induced increased expression of the adapter protein *MYD88* involved in TLR signaling.

Finally, we compared gene expression between all the study groups for the selected genes identified from Figure 4C–E. Fecal supernatants from UC patients induced higher colonoid gene expression of *IL12A* (Figure 5A), and tended to reduce *CCL2* (Figure 5B, *p* = 0.15), when compared to stimulation with supernatants from healthy subjects. Furthermore, fecal supernatants from UC patients induced higher expression of *IL10RA1* (Figure 5C) when compared to IBS patients, and lower expression of *CCL2* (Figure 5B) when compared to IBS patients. The addition of fecal supernatants from CC patients induced higher expression of *IFNA1* (Figure 5D) and *TRAF6* (Figure 5E), when compared to IBS patients. In summary, the addition of fecal supernatants to apical-out colonoids induced distinct gene expression patterns, which differ between the patient groups.

## 3. Discussion

In this study, the metabolite compositions of fecal supernatants differed between patients with CC, UC and IBS and altered the gene expression profiles when cultured with Caco-2 cells and colon-derived organoids. Hence, in vitro cell culture models, cultured in the presence of fecal supernatants to simulate interactions between the intestinal epithelium and the luminal content, pose a promising approach to study the gut barrier crosstalk in intestinal diseases.

During the past decade, studies that have investigated the relationship between intestinal diseases and gut microbiota deviations have escalated, particularly those reporting impaired microbial diversity [24,25,26,27,28]. The evidence supports the complex nature of these conditions that involve shifts in the microbiota composition and its metabolism, intestinal epithelial barrier dysfunction and inappropriate immune response, in intertwined with self-feeding connections that culminate in chronic inflammation [29]. Current strategies to investigate the convolute relationship between the local microenvironment, immune response and barrier integrity have focused on the characteristics of the immune response and microbiota composition separately. For this reason, there is a need for in vitro set-ups, evaluating the effects of the luminal content in the context of disease on cell cultures, to understand the participation of the diseased luminal environment in intestinal epithelium homeostasis. In this study, we exposed Caco-2 cell monolayers and colon-derived organoids from a single healthy donor to patient-derived fecal supernatants to determine the effects of the luminal content on the gene expression patterns of the intestinal epithelium. The fecal supernatants from patients with CC, UC and IBS altered the gene expression profiles of Caco-2 cells and organoids, although the gene regulation differed between the two in vitro models. However, the effect of fecal supernatants cannot be expected to be the same in the two model systems, as they differ in structure and composition of cells. Nevertheless, our results support the hypothesis that fecal supernatants represent the luminal microenvironment of patients with different gastrointestinal diseases and induce specific effects on the intestinal epithelial barrier.

Gut metabolites jointly reflect diet intake, modified human metabolites, microbial metabolism and environmental chemical compounds, all of which shape the barrier function [30]. They are, therefore, a dynamic imprint of the present luminal environment, hosts’ habits and a translation of the microbial activity rather than diversity fluctuations, as observed in studies that focus on microbiota. Accordingly, an untargeted metabolite analysis revealed that the fecal metabolite composition patterns differed between the patient groups and healthy subjects in the present study. Although our chosen method for analyzing metabolites did not allow us to annotate the compounds to identify any potential disease-specific biomarkers for further validation, these results suggest that the fecal metabolite composition may serve as a non-invasive diagnostic tool for gastrointestinal diseases in the future. Indeed, differences in the metabolite composition of patient-derived fecal samples in IBS, UC and CC patients when compared to healthy subjects have been reported in the past, supporting our findings [31,32,33]. Thus, our study indicates that the luminal content is disease-specific, and the effects observed in the cell cultures simulate in vivo events.

A strength of our study is the use of two different cell line set-ups for evaluating the effects of fecal supernatants on the epithelial barrier. The human-derived colon carcinoma cell line, Caco-2, is the most commonly used cell line to model the intestinal epithelium, due to its large availability and relatively simple maintenance. Transwell^®^ culturing techniques have long been used to induce the differentiation of Caco-2 cells into typical polarized IECs and facilitate in vitro access of both apical and basolateral compartments for the study of the effects of different stimuli on the intestinal epithelium [34]. Indeed, the effects of stimulation with fecal supernatants were observed in Caco-2 monolayers and were particularly distinct from single stimuli with LPS or TNF. Luminal metabolites, such as short chain fatty acids and polyamines, have been shown to support cell metabolism and improve barrier function [35,36]. Thus, changes in cell metabolism, measured by the MTT assay, may indicate that the luminal environment from a particular study group either harbors harmful content or lacks important metabolites that support cell function. Interestingly, the addition of fecal supernatants from healthy subjects in higher concentrations appeared to improve Caco-2 cell metabolism when compared to the control stimulations, which is in line with the aforementioned positive effects of the luminal metabolites on the gut barrier [35,36]. However, the Caco-2 cells do not reflect the heterogeneity of the intestinal epithelium, as they only differentiate into enterocyte-like cells and lack other cell types that are normally present in the epithelial layer [37]. Although the Caco-2 cell line is valid as a model for studying the properties of enterocytes following specific stimuli, as performed in this study, culture-related conditions related to passages may hamper our ability to generalize and compare results in the literature [38]. Furthermore, there is evidence of the substantial heterogeneity of sub-populations of Caco-2 cells, where some are parental lines passaged from the original tumor, whereas others are clonal lines selected to express specific characteristics and passaged further [38].

Stimulation of Caco-2 monolayers with fecal supernatants from the patient groups seemed to induce responses that suggest barrier disruption, such as increased expression of the tight junction-associated protein *CLND2* and apoptosis inhibitor *BIRC3*. Upregulation of *CLDN2* has been reported as an important response to diarrhea [39,40]. Another consistent feature of Caco-2 cells stimulated with fecal supernatants from the patient groups was the reduced expression of CD1d [41], which is prominently expressed in intestinal epithelial cells and is thought to regulate intestinal intraepithelial lymphocyte activity. As the Caco-2 monolayers only comprise epithelial cells, it could indeed be expected that the major effect of fecal supernatants in this experimental set-up is related to their ability to maintain and regulate the integrity of the epithelial cell layer.

The most prominent effects when stimulating colonoids with patient fecal supernatants were related to cytokine secretion, with an increase in *IL12A* and *TNF* gene expression. Indeed, IL-12a has been proposed as a key mediator for initiating colitis in animal models, triggered by epithelial barrier dysfunction and interaction with the gut microbiota [42,43]. Furthermore, increased expression of *TNF* in colonoids treated with fecal supernatants from CC and UC patients was also observed. While in CC patients, TNF has been associated with DNA damage and carcinogenesis [44], TNF secretion by IECs in murine models of colitis has been shown to participate in the initiation of chronic inflammation [45], suggesting that epithelial barrier responses are crucial for disease pathogenesis. Thus, these results indicate that an increase in *IL12A* and *TNF* may reflect in vivo immune activity triggered by luminal factors in disease settings.

There are similarities and differences between the two models studied. When compared to fecal supernatants from healthy subjects, the different patient groups primarily affected barrier integrity in the Caco-2 model, whereas gene expression of proinflammatory genes was the primary target in the organoid cultures. Fecal metabolites, particularly short chain fatty acids, have been shown to contribute to inflammatory responses through several mechanisms, including reduction in NFkB activation by bacterial components and improvement of barrier integrity by serving as an energy source to IECs, up-regulation of tight junction proteins and induction of mucin secretion. An imbalance in the fecal metabolite composition has been reported in CC, UC and IBS patients in comparison to healthy subjects [46,47,48], which may explain the regulation of tight junctions and pro-inflammatory patterns observed in the in vitro systems when stimulated by fecal supernatants from patients in this study. In Caco-2 monolayers, the expression of *BCL2, STAT3* and *NFKB1* was decreased, while *TNF* showed higher expression in colonoids, when treated with fecal supernatants from CC patients. Critical events during intestinal tumorigenesis are extensively described in the literature; however, the role of the microbiota is yet to be explained. Epithelial dysplasia has been reported to be triggered by increased *BCL2* expression via TLR sensing and NFkB activation in the presence of microbial imbalance [49]. Furthermore, NFkB activity may sustain cell survival by the activation of the oncogenic transcription factor Stat3 via IL-6 and TNF secretion [50]. Thus, luminal content, including microbial-derived components, present in fecal supernatants from CC patients potentially affect survival signals in cells of the epithelial layer.

Human-derived colon organoids cultured from crypt pluripotent stem cells may provide a more complete model than Caco-2 cells to study gut barrier interactions with the luminal environment [12]. In our study, disease-specific effects of fecal supernatants on colonoids were also observed, although they were different from the effects observed for the Caco-2 monolayers. These differences are likely to be explained by the fact that the colonoids host other specialized cells, such as goblet cells and enteroendocrine cells, and may therefore better reflect the multiplex cell–cell dynamics of the epithelium and how it is affected by fecal supernatants. Furthermore, colonoids are less influenced by the passaging effect as compared to Caco-2 cells, which derive from decades of culture passages, affecting metabolic cell activities, tight junction expression and possible cell responses [38]. In summary, different results are obtained for the two in vitro models, but the alterations observed in the colonoid system possibly simulate physiological events better than the Caco-2 monolayer set-up.

In this study, the use of apical-out colonoids limited the access to the basolateral compartment, as it was enclosed in the center of the colonoid. Thus, this experimental set-up did not allow for TNF to be used as a positive control stimulus or in TEER measurements. In further support of the findings in this report, our group have recently demonstrated that monolayers of healthy-derived colonoids, comprising a multicellular phenotype, also express a distinct gene profile when stimulated with fecal supernatants from IBS patients [21]. Taken together, both studies provide a proof of concept that fecal supernatants constitute a proxy for disease-specific metabolite composition and alter the gene expression profiles of intestinal cell lines.

We acknowledge that this study has limitations. The low number of samples for each group restricts the statistical power and small, but still physiologically relevant, differences between groups may not have been detected. Furthermore, the patients provided one fecal sample each, thereby only representing a single time-point, and intra-patient variability was not evaluated. However, in our work (manuscript in preparation), as well as in the work of other authors [51], we observed that the gut microbiota and fecal metabolome are stable over time (hours, days and weeks) within an individual, irrespective of stool consistency or frequency [52,53,54]. Apart from microbiota-derived metabolites, human-derived metabolites may also influence the fecal metabolite profiles. However, this aspect was not addressed. Separating these two entities from one another is a very cumbersome and delicate task, perhaps not even possible to achieve, and falls outside the scope of this manuscript, but deserves future investigations. Concerning the patients included in this study, clinical information of the potential pharmacological treatments was limited, which may introduce bias to the fecal metabolic composition and the effects observed in the in vitro systems. Furthermore, neither of the experimental set-ups allow long-term culture of the cells with fecal supernatants after differentiation, as in both cases, the cells start decaying. Future studies evaluating the long-term effects of fecal supernatants on cell growth, proliferation and secretion of proteins are necessary to fully understand the effects of luminal stimuli on cell culture systems of the intestinal epithelial barrier. Regarding the organoid model, we have used colonoids derived from one single healthy donor and our results may be influenced by the donor’s individual genetic characteristics; therefore, studies comparing different donors are warranted for further validation. Comparing the gene expression of fecal supernatant-stimulated cell lines to that of colonic mucosal biopsies patients and healthy donors would have added legitimacy to our experimental approach, as would an unbiased RNA sequencing analysis of gene expression. Furthermore, the choice of studying colonoids cultured as apical-outs over monolayers, to overcome the issue with obtaining confluence in the latter, did not allow us to access the basolateral compartment for stimulation or assessing cytokine secretion. However, the relative simplicity of the apical-out system, as compared to monolayers, enables future replication of our experimental set-up.

In conclusion, stimulation of Caco-2 cell monolayers and colon organoids with fecal supernatants derived from CC, UC and IBS patients alter the gene expression profiles, potentially reflecting the luminal microenvironment of the fecal sample donor. Therefore, we propose that the experimental approach described in this study provides a proof of concept for future studies that investigate the crosstalk between the microbiome and the gut barrier, in order to gain a better understanding of the effects of the environment in the pathogenesis of intestinal diseases.

## 4. Materials and Methods

### 4.1. Study Subjects

Adult study subjects (>18 years old) with histologically diagnosed CC awaiting surgery, histologically verified diagnosis of UC, IBS patients that met the Rome III criteria [55], and healthy subjects were recruited at the Sahlgrenska University hospital, Gothenburg, Sweden. The patients with newly diagnosed CC had fecal samples collected immediately prior to tumor resection surgery and were later staged according to the TNM international classification [56]. The patients with UC had fecal samples collected during a disease flareup, which was defined as an endoscopic Mayo score of ≥2 and a total Mayo score of ≥3, according to international criteria [57]. Patients with IBS were assessed for symptom severity using the IBS Severity Scoring System (IBS-SSS) [58] and were classified into IBS subtypes according to the evaluation of a 2-week stool diary using the Bristol Stool Form scale [55,59]. For this study, only patients subclassified as having predominantly diarrhea (IBS-D) or mixed bowel habits (IBS-M) [55] with moderate to severe IBS symptoms (IBS-SSS score of >175 points) were included [58].

Healthy subjects had no current or prior history of gastrointestinal or other chronic disorders, nor had they taken any medication, including immunosuppressive agents, during the 3 months before sample collection, and they reported no current GI symptoms.

### 4.2. Fecal Samples and Fecal Supernatant Preparation

Fecal supernatants extracted from fecal samples were used for metabolomics analysis and stimulation of Caco-2 cell monolayers and colon-derived organoids. Patients with UC or IBS and healthy subjects had their fecal samples collected at home and stored at −80 °C in the research facility, until preparation of fecal supernatants. Fecal samples from CC patients were collected following surgery, snap-frozen in liquid nitrogen and stored at −80 °C at the research facility, until the preparation of fecal supernatants. Feces were mixed with 2 weight volumes of ice-cold PBS, followed by centrifugation for 10 min at 3000× *g*. The resulting supernatant was ultra-centrifuged at 35,000× *g* for 2 h, at 4 °C. The collected fecal supernatants were stored in aliquots at −80 °C until further use.

### 4.3. Liquid Chromatography–Mass Spectrometry

Metabolites of the fecal supernatants were analyzed at Chalmers Mass Spectrometry Infrastructure (Gothenburg, Sweden), using a non-targeted liquid chromatography–mass spectrometry (LC–MS) approach. Samples were analyzed as single samples. Sample preparation and analysis are fully described in the Appendix A. Briefly, the analyses were carried out using reversed-phase chromatography and hydrophilic interaction chromatography with both positive and negative electrospray ionization [60]. The samples of each study group were analyzed in separate batches, which included their own quality control samples. The analytical workflow named “notame”, described by Zheng et al. [60], was used to pre-process the acquired data and included drift correction within and between the batches. Data imputation was performed using the *missForest* R package [61] and clustering of the features was performed to remove weak and repeated features [60]. Log_10_ transformation was applied prior to between-batch correction to reduce possible batch effects caused by the instrument.

### 4.4. Caco-2 Cell Culture and Stimulation with Fecal Supernatants

The Caco-2 cells, a cell line derived from human colon carcinoma, used in these experiments were originally obtained from Dr. Åsa Keita, Linköping, Sweden. Cells were initially cultured in T25 flasks (Corning^®^, Sigma, Søborg, Denmark) for expansion at 37 °C and 5% CO_2_ in cell culture medium, composed of high glucose Dulbecco’s Modified Eagle Medium (DMEM; Gibco^®^, Life Technologies^TM^, Carlsbad, CA, USA), supplemented with 5% AB human serum (Sigma-Aldrich, St. Louis, MO, USA), 2% L-glutamine (Gibco), 1% penicillin-streptomycin (Gibco) and 1% non-essential amino acid solution (Gibco). Cells at passage 23 were seeded at a density of 1.5 × 10^5^ cells/cm^2^ in the apical chambers of a 24-well Transwell^®^ clear 0.4μm pore polyester membrane insert (Corning^®^). Cell culture medium was changed in the apical and basolateral compartments every other day for 20 days, until full polarization and differentiation of the Caco-2 cell monolayer was achieved. Monolayer integrity and stabilization were evaluated by transepithelial electrical resistance (TEER; Millicell ERS-2 Voltohmeter, Merck, Darmstadt, Germany) every 4 days. Blank Transwells^®^ with no seeded cells were kept with cell culture medium for measuring the background resistance conferred by the membrane. TEER values are expressed as resistance normalized to area (resistivity) and were calculated by subtracting the background resistance from the resistance obtained in the monolayers and multiplying this value by the membrane surface area.

After 20 days of cell culture in Transwells^®^, cells were cultured for 48 h with medium that contained fecal supernatants from the different study groups diluted at 1:100, which were added to the apical chamber. For untreated controls, equal amounts of complete medium were added to the wells. Monolayers stimulated with lipopolysaccharides (LPS, Invivogen, San Diego, CA, USA) at 100 ng/mL/well, added to the apical side and tumor necrosis factor (TNF; Sigma-Aldrich, St. Louis, MO, USA) at a final concentration of 10 ng/mL/well added to the basolateral side, were included as inflammatory controls. TEER measurements were performed immediately before adding cell culture medium with or without fecal supernatants, LPS or TNF and after 48 h of stimulation. Following TEER measurement at 48 h, cells were collected in RA1 lysis buffer (Nucleospin^®^ RNA XS purification kit, Macherey-Nagel^TM^, Düren, Germany) and stored at −80 °C for gene expression analysis.

### 4.5. MTT (Tetrazolium) Assay

The toxic effect of the fecal supernatants on Caco-2 cells in vitro was assessed using the MTT assay, which measures cell metabolic activity as an indirect measurement of cell toxicity [62,63]. Caco-2 cells were seeded in 96-well, flat-bottom plates (Corning) at a concentration of 62,000 cells/cm^2^, and pre-incubated for 24 h in a 37 °C, 5% CO_2_ incubator before stimulation with fecal supernatants. Next, the cell culture medium was replaced by medium that contained fecal supernatants from three healthy donors in five dilutions (1:50, 1:100, 1:200 and 1:1000), and further incubated for 48 h. Untreated wells (controls) were kept for subsequent calculation of metabolic activity. After 48 h of incubation with fecal supernatants, cell viability assay was performed using the Cell Proliferation Kit I (MTT; Sigma Aldrich, Mannheim, Germany), according to the manufacturer’s protocol. Absorbance was measured using a microplate reader. Metabolic activity is presented as percentages (sample absorbance/average absorbance of control wells).

### 4.6. Generation of “Apical-Out” Colonoids and Stimulation with Fecal Supernatants

Following culture expansion, as fully described in the Appendix A, colonoids at passage 12, cultured in Matrigel and following the formation of 3D structures with the apical side facing inwards (apical-in), were prepared for eversion into “apical-outs” (apical side facing outwards to the medium), as described by Co and colleagues [12]. Briefly, the Matrigel was dissolved by incubating the plates on ice with Cell Recovery Solution (Corning). After dissolution of the Matrigel, the well contents were collected into a conical tube and centrifuged at 1100 rpm for 7 min. Next, the supernatant that contained the Matrigel and Cell Recovery solution was discarded, and the colonoids were carefully re-suspended in differentiation medium that contained IntestiCult^TM^ OGM Human Component A, 10 mM HEPES and Primocin diluted to 1:500. Lastly, the colonoids were distributed in 24-well low attachment plates (Corning^®^, Corning, NY, USA), at a concentration of approximately 100–150 colonoids per well, and incubated for 72 h at 37 °C and 5% CO_2_. The withdrawal from the critical growth factors present in IntestiCult^TM^ OGM Human Component B allowed the differentiation of the crypt stem cells into various intestinal cell subtypes. Furthermore, the absence of Matrigel as a basement matrix evokes an inversion of the organoids’ polarization, causing the apical side to face outwards [12,64].

After 72 h, the apical-out colonoids were split into 96-well plates (~80 colonoids per well) and cultured in differentiation media that contained fecal supernatants from each of the study subjects at a dilution of 1:100. For untreated controls, equal amounts of advanced DMEM were added to the cell cultures. Colonoids stimulated with lipopolysaccharides (LPS, Invivogen), diluted in differentiation media at a final concentration of 100 ng/mL, were included as inflammatory controls. After 48 h of culture, the colonoids were collected in RA1 lysis buffer (Nucleospin^®^ RNA XS purification kit) and stored at −80 °C for gene expression analysis.

### 4.7. RNA Extraction and cDNA Preparation

Total mRNA was extracted using the Nucleospin^®^ RNA XS purification kit, according to the manufacturer’s protocol. Measurement of mRNA concentration was performed using a NanoDrop ND-1000 (NanoDrop Technologies, Wilmington, DE, USA). Complementary DNA (cDNA) was prepared using the RT² First Strand Kit (Qiagen, Hilden, Germany), according to the manufacturer’s protocol.

### 4.8. RT^2^ Profiler PCR Array

A custom RT² PCR Array (Qiagen) was analyzed on a Quantstudio 12K Flex Real-Time PCR System (Applied Biosystems, Foster City, CA, USA), using the RT² qPCR SYBR Green ROX MasterMix (Qiagen). The gene expression custom array comprised 96 genes (Appendix A), including 5 housekeeping (HK) genes (GAPDH, HPRT1, RPLP0, ACTB; B2M) and 4 quality controls (positive PCR control, reverse transcription control, genomic DNA control; negative template control). All samples passed the quality controls. The missing values and CT values > 38 were set to 38. Genes were excluded from the analysis if >50% of the samples had missing or very high CT values (CT > 36). In total, 18 genes were excluded from the Caco-2 cell analysis, and 19 genes were excluded from the colonoid analysis. Normalized values are expressed as 2^-(CT Target gene-mean CT of HK genes)^. Samples were run as single samples, according to the company’s recommendation and our in-house experience with the assay.

### 4.9. Statistical Analyses

Principal component analysis (PCA) was conducted on LC–MS and gene expression data using the prcomp-algorithm and visualized using the pca3d-package in R software (version 3.3.2; R Core Team). Orthogonal partial least squares discriminant analyses (OPLS-DA) were implemented to correlate the patient groups with gene expression data in the linear multivariate models using the SIMCA software (version 16; MKS Data Analytics Solutions, Umeå, Sweden). The R2Y parameter represents the goodness of fit of the model (best possible fit: R2Y = 1). The Q2 parameter represents an estimate of the predictive ability calculated by cross-validation (best possible predictive ability: Q2 = 1). For heterogeneous biological variables, a model is considered to have a good fit with an R2Y ≥ 0.5 and a good predictive ability with a Q2 > 0.4. A variable influence on the projection (VIP) cutoff was defined based on the discriminatory power of the model, and the cutoff that best explained the y observations, based on R2Y and Q2 values, was selected. In the OPLS-DA loading column plots, each x-variable is shown in relation to y and the importance of each x-variable is represented by column bars. The x-variables positioned furthest to the left or right are the most closely related to the respective y-variable, and thus contribute the most to the model. The larger the bar and the smaller the 95% confidence interval (shown by the whiskers), the stronger and more reliable the contribution is to the model.

The differences between two groups were assessed using the Mann–Whitney U test and the differences between three or more groups were assessed using the Kruskal–Wallis test, followed by Dunn’s multiple comparison test. For TEER analysis data, differences between the baseline and 48 h measurements were assessed using the paired sample t-test. Statistical analyses were performed using GraphPad Prism (GraphPad Software version 9); *p*-values < 0.05 were considered as statistically significant. The demographic data are shown as the median values (range), and gene expression data are presented as the median values (interquartile range).

## Figures and Tables

**Figure 1 ijms-23-15505-f001:**
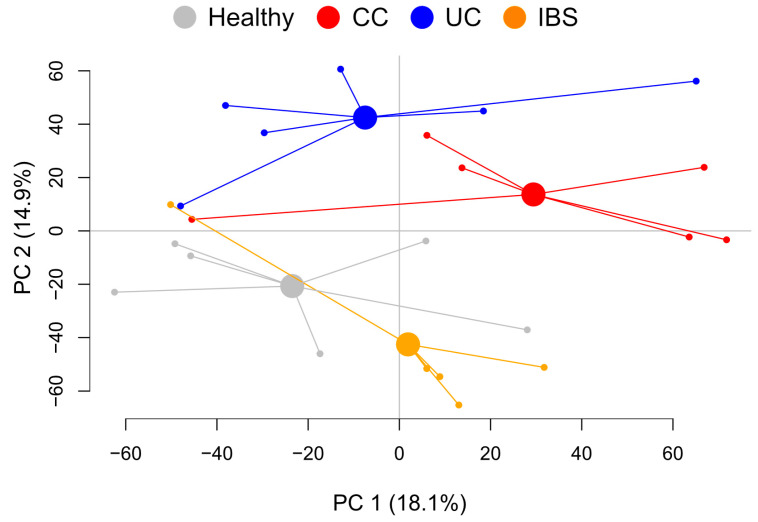
Metabolite profiles of fecal supernatants from healthy subjects and patients with colon cancer (CC), ulcerative colitis (UC) and irritable bowel syndrome (IBS). Fecal supernatants were analyzed by untargeted LC/MS. Results for the detected 9,699 spectral features of the fecal supernatants are shown for principal component analysis (PCA). *n* = 6 in each group. CC, colon cancer; UC, ulcerative colitis; IBS, irritable bowel syndrome; PC, principal component.

**Figure 2 ijms-23-15505-f002:**
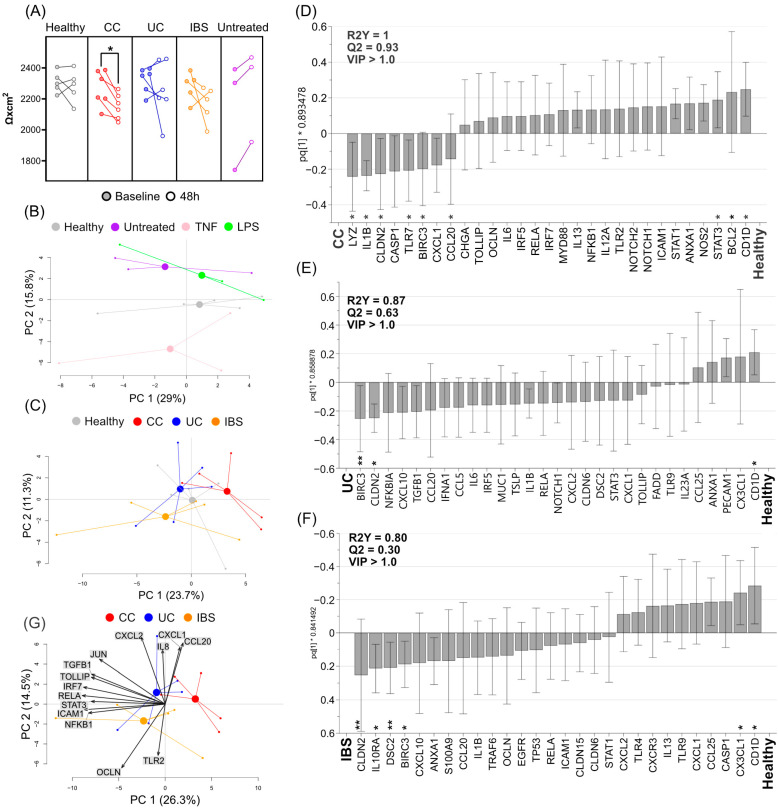
Gene expression of Caco-2 monolayers stimulated with fecal supernatants from healthy subjects and patients with colon cancer (CC), ulcerative colitis (UC) and irritable bowel syndrome (IBS). Caco-2 monolayers were differentiated for 20 days and stimulated apically with fecal supernatants and LPS or basolaterally with TNF or left untreated (cell culture medium alone), for 48 h. (**A**) Transepithelial electrical resistance was measured before (filled dots, each dot representing one sample) and after stimulation (corresponding unfilled dot), for healthy (grey), CC (red), UC (blue) and IBS (orange). Gene expression was analyzed by a custom PCR array. (**B**) Principal component analysis (PCA) for cells treated with fecal supernatants from healthy subjects (grey), LPS (green), TNF (pink) and media (purple). (**C**) PCA for cells treated with fecal supernatants from healthy subjects and CC (red), UC (blue) and IBS (orange) patients. Column loading plots from orthogonal partial least squares discriminant analyses (OPLS-DA) using a VIP cut-off > 1.0, comparing (**D**) healthy subjects and patients with CC, (**E**) healthy subjects and patients with UC and (**F**) healthy subjects and patients with IBS. (**G**) A PCA biplot for cells treated with fecal supernatants from patients with CC, UC and IBS. Between-group comparisons were tested with Mann–Whitney U test; asterisks identify statistically significant p values: * *p* < 0.05; ** *p* < 0.01. LPS (*n* = 3), TNF (*n* = 3), media (*n* = 3); fecal supernatants (*n* = 5 for each group). CC, colon cancer; UC, ulcerative colitis; IBS, irritable bowel syndrome; TNF, tumor necrosis factor; LPS, lipopolysaccharide; PC, principal component.

**Figure 3 ijms-23-15505-f003:**
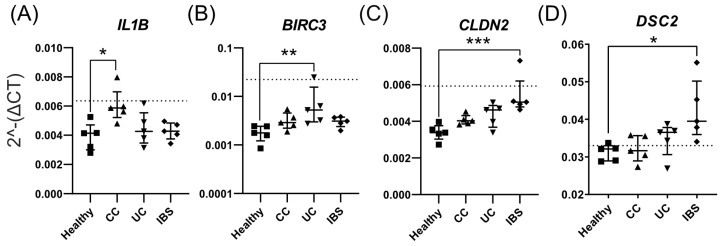
Comparison of gene expression of Caco-2 monolayers stimulated with fecal supernatants from healthy subjects and patients with colon cancer (CC), ulcerative colitis (UC) and irritable bowel syndrome (IBS). Caco-2 monolayers were differentiated for 20 days and stimulated apically with fecal supernatants or left untreated (medium alone) for 48 h. Gene expression of (**A**) *IL1B* (**B**) *BIRC3*, (**C**) *CLDN2* and (**D**) *DSC2* was compared between healthy (square), CC (triangle), UC (inverted triangle) and IBS (rhombus). The dotted line indicates the mean gene expression of untreated Caco-2 cells *(n* = 3). Statistical difference between the groups was assessed by the Kruskal–Wallis test, followed by Dunn’s multiple comparisons test. * *p* < 0.05; ** *p* < 0.01; *** *p* < 0.001 (*n* = 5 for each group).

**Figure 4 ijms-23-15505-f004:**
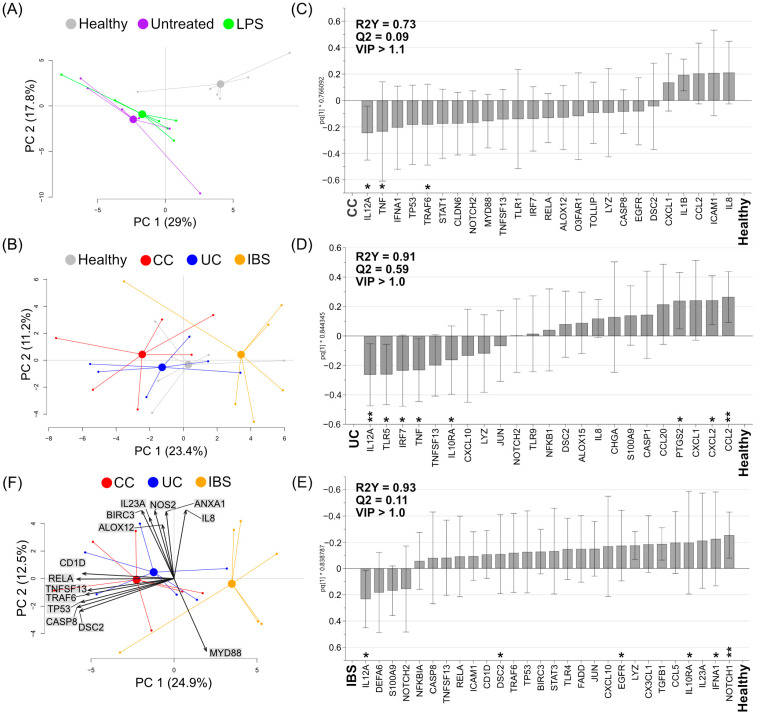
Gene expression of colonoids stimulated with fecal supernatants from healthy subjects and patients with colon cancer (CC), ulcerative colitis (UC) and irritable bowel syndrome (IBS). Apical-out colonoids were stimulated with fecal supernatants, LPS or left untreated (media) for 48 h. Gene expression was analyzed by a custom PCR array. (**A**) Principal component analysis (PCA) for colonoids treated with fecal supernatants from healthy subjects (grey), LPS (green), and media (purple). (**B**) PCA for colonoids treated with fecal supernatants from healthy subjects (grey) and CC (red), UC (blue) and IBS (orange) patients. Column loading plots from orthogonal partial least squares discriminant analyses (OPLS-DA) using a VIP cut-off > 1.0 or >1.1, comparing (**C**) healthy subjects and patients with CC, (**D**) healthy subjects and patients with UC and (**E**) healthy subjects and patients with IBS. (**F**) A PCA biplot for colonoids stimulated with fecal supernatants from CC (red), UC (blue) and IBS (orange) patients. Between-group comparisons were tested with Mann–Whitney U test; asterisks identify statistically significant p values: * *p* < 0.05; ** *p* < 0.01. LPS *(n* = 6), media *(n* = 6); fecal supernatants *(n* = 6 for each group). LPS, lipopolysaccharide; CC, colon cancer; UC, ulcerative colitis; IBS, irritable bowel syndrome; PC, principal component.

**Figure 5 ijms-23-15505-f005:**
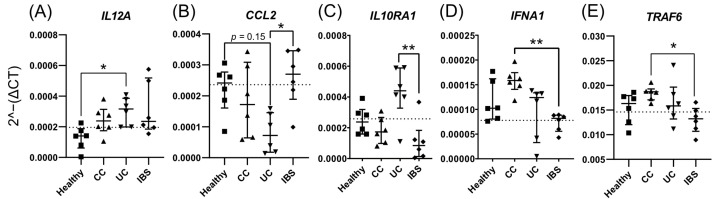
Comparison of gene expression of colonoids stimulated with fecal supernatants from healthy subjects and patients with colon cancer (CC), ulcerative colitis (UC) and irritable bowel syndrome (IBS). Apical-out colonoids were stimulated with fecal supernatants or media for 48 h. Gene expression of (**A**) *IL12A*, (**B**) *CCL2*, (**C**) *IL10RA1*, (**D**) *IFNA1* and (**E**) *TRAF6* was compared between healthy (square), CC (triangle), UC (inverted triangle) and IBS (rhombus). The dotted line indicates mean gene expression of untreated colonoids *(n* = 6). Statistical difference between the groups was assessed by the Kruskal–Wallis test, followed by Dunn’s multiple comparisons test. * *p* < 0.05; ** *p* < 0.01 *(n* = 6 for each group).

## Data Availability

The datasets used in this publication are available at https://figshare.com/articles/dataset/LC-MS_and_gene_expression_xlsx/20657694/1 (published 26 August 2022).

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
