# Peer review of "Fecal Luminal Factors from Patients with Gastrointestinal Diseases Alter Gene Expression Profiles in Caco-2 Cells and Colonoids"

_ijms, 2022, doi:10.3390/ijms232415505_

Round 1

Reviewer 1 Report

1.       The groups are too small (less than 10 samples) to speak of unique metabolic profiles. Moreover, human-derived metabolites may influence the data. What about biological replicates and technical controls?

2.       Why is cell viability greater than 100% in Figure S2? 1.  Please check the figure.

3.       How do you explain that the untreated controls are less viable than treated in Figure S2?

4.       What about data on the viability of Caco-2 after treatment with LPS and TNF? Is there any changes?

5.       Did you perform biological replicates for gene expression analysis?

6.       Whether a comparative analysis of gene expression of Caco-2 monolayers was carried out between groups CC, UC and IBS?

7.       How LPS and TNF affect gene expression of Caco-2 monolayer and colonoids? No any data on gene expression in the results.

Author Response

We thank you for the thorough review and the comments given on the manuscript “Fecal luminal factors from patients with gastrointestinal diseases alter gene expression profiles in Caco-2 cells and colonoids”

Please find our point-to point response to your comments below. The manuscript has been revised accordingly, and changes in the main document have been highlighted in yellow.

Yours sincerely

Luiza Moraes Holst PhD

Department of Microbiology and Immunology

University of Gothenburg

luiza.moraes.holst@gu.se

  1. The groups are too small (less than 10 samples) to speak of unique metabolic profiles. Moreover, human-derived metabolites may influence the data. What about biological replicates and technical controls?

We agree with the reviewer that the patient groups are too small to identify unique metabolites distinguishing different patient groups from one another or from healthy subjects. However, the minor overlap between overall metabolite profiles of the study groups suggest that they do indeed differ. However, as implied by the reviewer, due to the sheer size of the study we cannot claim that these profiles are unique for the gastrointestinal diseases studied. We have therefore modified the discussion on page 7 lines 254-256 and page 8 lines 285-289. We have also included the aspect that human-derived metabolites may influence the metabolite profiles as a potential weakness of the study on page 10 lines 387-393. However, to identify and separate microbial-derived from human-derived is a very cumbersome task, perhaps not even possible to achieve, and falls outside the scope of this manuscript.

With regards to analysis of the fecal metabolome with LC-MS, six biological replicates were used (samples from 6 different subjects per group) but no technical replicates of samples were run, i.e., samples were not run in duplicate/triplicate. However, our previous method development has demonstrated that different aliquots from the same fecal supernatant give rise to comparable results. Further, different preparations of fecal supernatants from the same fecal sample give rise to comparable metabolomic profile. Information related to our in-house experiences with LC-MS has been added to the methods section on page 11 line 450. In addition, we are currently preparing another manuscript, demonstrating that the fecal metabolome profile is stable over time (hours, days and weeks) within an individual, which has been added to the discussion on page 10 lines 386-393.

  1. Why is cell viability greater than 100% in Figure S2? 1. Please check the figure.

We thank the reviewer for the question, the data shown are indeed correct. See further comments concerning “viability” in point 3 below.

  1. How do you explain that the untreated controls are less viable than treated in Figure S2?

We apologize for an error in Figure S2.  The figure title should state metabolic activity, not cell viability. The MTT assay can be used as an indirect measurement of cell viability and could reflect cell death if the activity is decreased in relation to control, however we detected the opposite. The untreated controls are Caco2 cells cultured without the addition of fecal supernatants, containing a plethora of metabolites and constitute nourishment for the cell cultures. Thus, the addition of fecal supernatant to the cell culture media does provide a more physiological condition for the Caco-2 cells, replicating the intestinal environment, thereby allowing them to thrive more than the untreated controls. By using this method, we cannot define if the cells have divided more or if they only increased their metabolism. Due to this we have removed the comment concerning viability in the discussion on page 8 lines 308-310 “Interestingly, addition of fecal supernatants from healthy subjects in higher concentrations appeared to improve Caco-2 cell metabolism and viability when compared to the control stimulations…”.

  1. What about data on the viability of Caco-2 after treatment with LPS and TNF? Is there any changes?

We agree that this would have been interesting to evaluate but we did not analyze metabolic activity / viability of LPS and TNF treated Caco2 cells. The MTT assay was merely used to define an appropriate concentration of the fecal supernatants as stated in the results section.

  1. Did you perform biological replicates for gene expression analysis?

Our study included biological replicates, i.e., there were 5-6 study subjects in each study group. We also included positive (TNF a proinflammatory cytokine) and negative (no addition of fecal supernatant) controls as technical controls of the cell culture systems stimulated with fecal supernatants from the different study groups. However, we did not include technical replicates such as duplicates, as our long-term experience is that these gene expression arrays are stable and provide reliable results with single samples. Also, according to Qiagen their PCR array system demonstrates strong correlations across technical replicates (average correlation coefficients >0.99). These aspects of the study have been clarified on page 13 lines 546-548.

  1. Whether a comparative analysis of gene expression of Caco-2 monolayers was carried out between groups CC, UC and IBS?

Yes, we did indeed compare the gene expression of Caco-2 cells cultured in the presence of fecal supernatants. Figure 2G, a PCA biplot shows the genes most importantly for differentiating between the patient groups. Figure 3A-D shows the genes that statistically differed between the study groups. However, for more detailed analysis using OPLS-DA, we only chose to compare the different disease groups to healthy (Figure 2).

  1. How LPS and TNF affect gene expression of Caco-2 monolayer and colonoids? No any data on gene expression in the results.

We thank the reviewer for the question. The gene expression for TNF and LPS stimulated Caco-2 cells were analyzed and shown in a PCA in figure 2b. These cell cultures were included as controls of the system and compared to addition of fecal supernatants from healthy subjects and untreated cell cultures. We did not show the gene regulation of the controls in more detail due to limitation of space and interest of the manuscript, but it is clear from the PCA in figure 2B, that TNF and LPS had effect on the cell cultures. The results are explained on page 5 lines 152-154. Considering that only few cell culture wells were included in the TNF and LPS groups (n=3), as they were added as positive and negative controls, showing the gene expression of these analyses will provide limited information. However, we are of course willing to include these analyses as supplementary material if required by the editor/reviewers.

Reviewer 2 Report

Sorry for the late review. I read it and found it very interesting.

In this paper, you use organoids to show that the feces of patients with specific diseases can affect the epithelial barrier of the large intestine.

Please consider adding the following point.

result2.1 subject: It would be more interesting if there were a description of BMI, race composition, etc. This is because the normality of stools is expected to be significantly affected by them.

Author Response

We thank you for the thorough review and the comments given on the manuscript “Fecal luminal factors from patients with gastrointestinal diseases alter gene expression profiles in Caco-2 cells and colonoids

Please find our point-to point response to your comments below. The manuscript has been revised accordingly, and changes in the main document have been highlighted in yellow.

Yours sincerely

Luiza Moraes Holst PhD

Department of Microbiology and Immunology

University of Gothenburg

luiza.moraes.holst@gu.se

Please consider adding the following point; result2.1 subject: It would be more interesting if there were a description of BMI, race composition, etc. This is because the normality of stools is expected to be significantly affected by them.

We thank the reviewer for the thought-provoking comment. The large majority of study subjects were of Caucasian origin, but BMI was not recorded for the CC patients, and as these study subjects were substantially older than other study groups, we cannot exclude the possibility of them having a higher BMI. Still, the patients included in the study all had gastrointestinal disorders, with well-known effects on stool consistency and frequency. Thus, it is fair to assume that the normality of stool differed between patients and healthy subjects. However, as mentioned in comments to reviewer 1, we are currently preparing a manuscript demonstrating that the fecal metabolome profile is stable over time (hours, days and weeks) within an individual, and does not seem to be related to stool consistency or frequency. Therefore, it is most likely that the differences in metabolite compositions of fecal supernatants, and their effects on cell culture systems are related to disease specific factors and not merely to stool consistency or bowel habits. This has been included in the discussion on page 10 lines 386-393.

Round 2

Reviewer 1 Report

In my opinion, it is better to present data on patients in table format with clear indication of gender, stage of diagnosis, mean age with standard deviation etc. It will be much easier for readers to understand the studied groups.

Author Response

Comment: In my opinion, it is better to present data on patients in table format with clear indication of gender, stage of diagnosis, mean age with standard deviation etc. It will be much easier for readers to understand the studied groups.

Response: We appreciate the reviewer for this suggestion. We have now included a table depicting the cohort’s demographic and most relevant clinical information in the supplementary material (supplementary table 1). Furthermore, minor changes in the text have been made to match the inclusion of the table and skip repetitive information (results section, pages 2 and 3, lines 96-106). We thank once again the reviewer for the suggestions and believe these changes have improved the manuscript for publication.